# Effectiveness of a quality improvement intervention to increase adherence to key practices during female sterilization services in Chhattisgarh and Odisha states of India

Ashish Srivastava[1,2], Geeta Chhibber[1]*, Neeta Bhatnagar[3], Angela Nash-Mercado[3], Jyoti Samal[1], Bhagyashree Trivedi[1], Vinod Srivastava[1], Barbara Rawlins[3], Vivek Yadav[1], Bulbul Sood[1], Regien Biesma[2], Young-Mi Kim[3], Jelle Stekelenburg[2,4]

1 Jhpiego India, New Delhi, India, 2 Department of Health Sciences/Global Health, University of Groningen/ University Medical Center Groningen, Groningen, The Netherlands, 3 Jhpiego, Baltimore, Maryland, United States of America, 4 Department of Obstetrics and Gynecology, Leeuwarden Medical Center, Leeuwarden, The Netherlands

* Geeta.Chhibber@Jhpiego.org

## Abstract

### Background

In response to longstanding concerns around the quality of female sterilization services provided at public health facilities in India, the Government of India issued standards and quality assurance guidelines for female sterilization services in 2014. However, implementation remains a challenge. The Maternal and Child Survival Program rolled out a package of competency-based trainings, periodic mentoring, and easy-to-use job aids in parts of five states to increase service providers' adherence to key practices identified in the guidelines.

### Methods

The study employed a before-and-after quasi-experimental design with a matched comparison arm to examine the effect of the intervention on provider practices in two states: Odisha and Chhattisgarh. Direct observations of female sterilization services were conducted in selected public health facilities, using a checklist of 30 key practices, at two points in time. Changes in adherence to key practices from baseline to endline were compared at 12 intervention and 12 comparison facilities using a difference in difference analysis.

### Results

Several key practices were well-established prior to the intervention, with adherence levels over 90% at baseline, including hemoglobin and urine testing, use of sterile surgical gloves and instruments, and recommended surgical technique. However, adherence to many other practices was extremely low at baseline. The program significantly increased adherence to nine practices, including those related to ascertaining client's medical eligibility, client-provider interaction, the consent process, and post-operative care. The greatest improvement

**Data Availability Statement:** All relevant data are within the manuscript and its Supporting Information files.

**Funding:** The funding for this study was provided by the United States Agency for International Development (USAID) under the terms of the Cooperative Agreement AID-OAA-A-14-00028. The funders had no role in study design, data collection and analysis, decision to publish, or preparation of the manuscript.

**Competing interests:** The authors have declared that no competing interests exist.

was observed in the provision of written instructions for clients prior to discharge. At endline, however, adherence remained below 50% for 14 practices.

## Conclusion

Low adherence to key practices at baseline confirmed the need for quality improvement interventions in female sterilization services. While the intervention improved adherence to certain practices around admission and post-operative care, inadequate human resources and infrastructure, among other factors, may have blunted the impact of the intervention.

## Introduction

Female sterilization is the most popular contraceptive method globally: an estimated 190 million couples rely on the method, which constitutes 30% of the contraceptive method mix worldwide [1]. In India, female sterilization has dominated the contraceptive method mix since the early 1990s, as observed in multiple rounds of the Demographic and Health Surveys from 1991–92 to 2015–16 [2]. Each year nearly 3.5 million women receive sterilization services at public health facilities in India, and the procedure constitutes 67% of India's contraceptive method mix [2, 3].

Despite the wide use of female sterilization in India, poor quality of sterilization services in public health facilities has been a persistent concern, dating back to the early 1990s. Reported problems include, but are not limited to: inadequate client counselling on alternative long–acting methods, poor interpersonal interactions between service providers and clients, inadequate screening of clients for potential contraindications, poor maintenance of aseptic conditions during surgery, minimal monitoring of clients during and after surgery, and minimal written or verbal instructions offered to clients at discharge [4]. Government targets for female sterilization further exacerbated quality concerns as health workers felt pressured to meet locally imposed targets [5].

In response, Government of India (GoI) adopted a target-free approach in 1996 [5] and issued quality of care standards for female sterilization in 1998–99 [6]. Despite these efforts, evidence of the poor quality of female sterilization services in various states continued [7–10]. In 2005, the Supreme Court of India passed orders to improve the quality of female sterilization procedures. GoI incorporated these directives into the revised quality standards, which also established a quality assurance mechanism and a revised compensation scheme for adverse events, should they occur [11]. Although this led to some improvements, public sector female sterilization services still did not meet the prescribed quality standards [12, 13], resulting in an unacceptable number of deaths, complications, and failures [5, 14]. In 2014, a major shift in the discourse around female sterilization took place when 16 women died after undergoing sterilization surgery in a family planning camp in the state of Chhattisgarh. In the same year, the Supreme Court of India issued directives to make the program target-free and shift the focus from quantity to the quality of procedures [15]. This led to revisions in government guidelines to incorporate evidence-based best practices and quality assurance processes for female sterilization [16, 17]. While India's current national guidelines are comprehensive and address almost all components of quality in female sterilization services provided in the public health system, they still need to be understood within the context of implementation. Poor technical and managerial capacity, scarcity of trained human resources, and a high client load

have put increased pressure on the health system at the state level and below, resulting in underutilization of these guidelines [13].

Any assessment of female sterilization services must be cognizant of the global articulation of quality in the context of family planning. This emphasizes the need to both, maintain safety and incorporate client-centered care, although the latter is not usually accorded equal treatment [18]. The United Nations Committee on Economic, Social and Cultural Rights has defined quality as evidence-based practices that are scientifically and medically appropriate [19].

In light of the history of quality problems with female sterilization in India as well as the current focus on voluntary, client–centered quality family planning services [20, 21], the Maternal and Child Survival Program (MCSP) in India has worked to improve practices pertaining to female sterilization in public health facilities in five states: Assam, Chhattisgarh, Maharashtra, Odisha, and Telangana.

MCSP was a global program focused in 26 high priority-countries, which supported the Government of India in expanding the basket of contraceptive choices, contributing to meet India's FP 2020 commitments and to universal access to quality contraceptive services. Quality along with respectful client-centered care was the cornerstone of the program.

This study examines whether the program's package of interventions increased service providers' adherence to key practices during provision of female sterilization services at public health facilities in two of these states (Chhattisgarh and Odisha). The study was designed to inform program managers and other key stakeholders about critical practices requiring additional attention and resources. It also aimed at adding to the existing knowledge on the 'know-do gap'. While increased knowledge of health service providers is often seen as a process indicator of improved quality of care, it may not necessarily translate into change in practices in the real-world settings [22–24]. Providers may not 'do' as per what they 'know'. Therefore understanding the know-do gap is a critical step towards developing effective, practical strategies to improve delivery of quality female sterilization services.

## Methods

### Intervention package

**Clinical training.** Existing mini-laparotomy providers were identified in MCSP-supported facilities. During a two-day, hands-on refresher training, they were thoroughly oriented on a set of best practices outlined in the updated GoI guidelines on Standards & Quality Assurance in Sterilization Services (2014) as well as the Reference Manual for Female Sterilization [16, 17]. Providers' knowledge and skills were standardized on anatomic models on the first day of training, followed by clinical practice in the operating theater on the second day.

**Clinical safety checklist and supportive supervision.** Critical practices were also incorporated into a Clinical Safety Checklist (CSC). The CSC was inspired and informed by experience with the World Health Organization's (WHO) Surgical Safety checklist, which has proven to reduce adverse events in surgery [25]. The CSC was implemented at all MCSP- supported facilities and translated technical guidelines into an easy-to-understand job aid for use by the service providers responsible for delivering female sterilization services. It was also designed to increase the role of nurses, support teamwork, and serve as a quality improvement tool. The checklist, was organized around four 'pause points' that correspond to client flow at the facility from: (1) admission, (2) pre-operative assessment, (3) surgery, to (4) post-operative care and discharge.

The checklist was first introduced to a wide audience of administrators, facility managers, doctors and staff nurses through state- and district-level workshops. Facility-level orientations

on the tool and its standard operating procedures were then conducted for members of the core surgical team who were directly involved in providing sterilization services. The focus was on quality and each client's safety, overall experience, and satisfaction. During the first few months of implementation, project staff visited facilities on service days to support the surgical team by offering guidance, mentorship, and supportive supervision.

**Client card.** To strengthen reciprocal linkages between facility-based processes and the community, the program introduced a client card for female sterilization. Frontline workers (Accredited Social Health Activists or ASHAs) issued a client card to each woman when she first expressed a desire for adopting a permanent family planning method. Then community-based health workers (Auxiliary Nurse Midwives or ANMs) used the card to screen women's fitness for the sterilization procedure that included a blood pressure measurement, hemoglobin estimation, urine for proteins and sugar and ruling out pregnancy using the pregnancy checklist. ANMs also used the client card during follow-up visits after the surgery was performed. The card contained complete information for women on post-surgical "do's and don'ts" and key "to-do's" for follow-up visits. This 360-degree approach ensured that clients received quality care in the community before and after surgery, even as the quality of service provision was strengthened at the facility level.

## Study design and setting

The study employed a before-and-after quasi-experimental design with a matched comparison arm. It had a quasi-experimental design as the intervention districts for MCSP were selected purposefully and not randomly. The intervention districts were selected in consultation with state officials based on government priorities and poor facility performance on key family planning indicators.

The study was conducted in two of the five states where the MCSP intervention was rolled out–Chhattisgarh and Odisha–where all interventions were implemented with full intensity. To ensure uniformity of assessment, we conducted the study in health facilities where female sterilization procedures were done using the mini-laparotomy (Minilap) approach under sedation and local anesthesia; this was true of all facilities in Chhattisgarh and most facilities in Odisha.

The intervention was implemented in six districts in each state, for a total of 12 intervention districts. In intervention districts, MCSP supported all public health facilities which were designated by the government to provide Fixed Day Static (FDS) female sterilization services. The FDS approach offers regular sterilization services on fixed days throughout the year, performed by trained providers posted in that facility [16]. From among all MCSP-supported public health facilities that employed the minilap procedure in the 12 intervention districts, we randomly selected 12 facilities for the intervention group (3 of 44 facilities in Chhattisgarh and 9 of 108 facilities in Odisha). To form a comparison group, we matched the intervention facilities with 12 FDS facilities from nine districts that were not supported by MCSP, based on the facility's (1) state, (2) delivery volume, and (3) average monthly female sterilization client load during the three months prior to baseline data collection (Fig 1). At endline, three facilities from the comparison group were replaced, using the same matching criteria, because they no longer performed female sterilization by the Minilap approach.

## Sample size

For estimating the sample size, we used the following formula N = 2 $(Z_{\alpha/2} + Z_{\beta})^2$ P (1 –P) / $(p1 –p2)^2$, which is for comparison between two groups when the endpoint is qualitative [26]. When calculating the sample size, we assumed that health service providers adhered to 50% of

MCS program intervention in 152 facilities across 12 districts of Odisha and Chhattisgarh.

Random selection of 12 facilities (9 from Odisha and 3 from Chhattisgarh) to form the intervention group of study.

Selection of 12 matched facilities (on the basis of state, delivery load and monthly sterilization load) from 9 non-intervention districts to form the comparison group.

20 observations per pause point, per round of data collection at each facility. Total sample size of 960 observations: 240 observations per study arm per round of data collection

**Fig 1. Flow chart summarizing the selection of study sites and sample per study site.**

key female sterilization practices at baseline (p1), using 80% power (1 – β) to detect a 20% change (p1 –p2) in adherence at endline with 5% type I error (α). To account for clustering by health facilities, we inflated the sample size using a design effect of 2.5 (cluster size of 20, intra cluster correlation coefficient of 0.08). We rounded up the calculated sample size of 235 observations to 240 observations in order to evenly distribute them among 12 clusters (i.e., facilities) for each study group in each round. This resulted in a total sample size of 960 observations: 240 observations per study arm per round of data collection. These were evenly divided across facilities, so that 20 observations were made at each facility at baseline and again at endline.

Pause points served as the unit of observation rather than the individual client. Data collectors did not try to follow the same client through admission, pre-operative assessment, surgery, and discharge, which could be a lengthy process. Instead, they observed whichever client was available at a given pause point. As a result, any single client may have been observed at just one or two pause points. Thus, one complete observation (including all four pause points) may involve from one to three women (as pause points 2 & 3 were observed together), and the 20 observations at a given facility represent more than 20 women.

## Study participants

On reaching a facility, data collectors identified all health service providers who provided female sterilization-related services and invited them to participate in the study. These included doctors, nurses, and ANMs. All health service providers who were approached, agreed to participate and there were no refusals.

Women who underwent female sterilization procedures at the selected facilities during the two observation periods were eligible to participate in the study. All women who came for female sterilization on the given day were approached when they arrived at the facility, offered an explanation of the study, and invited to participate.

## Data collection tools and procedures

The study data came from direct observation of the provision of female sterilization services. Observations were recorded on a structured checklist (S1 Checklist) that included 30 key

practices outlined in national guidelines for female sterilization [16]. The practices were categorized into the same four pause points as the CSC. Some key practices were further subdivided into two to five observable steps (Table 1). For such practices, data collectors noted whether each step was performed or not performed. Adherence to a practice was defined as performing all steps in that practice.

Data collectors were medical doctors and nursing graduates, and 70% of those who collected data in the baseline round returned for the endline round. Data collectors attended a standardized three-day training prior to each round of data collection, during which the observation checklist and scoring procedures were reviewed. The training included practice sessions during which data collectors filled out the observation checklist while watching simulated procedures using anatomic models. Ensuing discussions of the observations helped standardize the application of the scoring system across data collectors and also clarified doubts and concerns.

Baseline data were collected in April-June 2017 and endline data in November-December 2018. During each survey round, two data collectors (one doctor and one nurse) were assigned to a facility. After obtaining consent from both, the providers and clients, data collectors observed interactions until the target number of observations were reached for each pause point. Nurses observed pause points 1 (admission) and 4 (post-operative care and discharge). Doctors observed pause points 2 (pre-operative assessment) and 3 (surgery).

## Statistical analysis

We computed the proportion of observations in which the provider adhered to a key practice and compared proportions at endline and baseline in each study arm by performing Chi square tests. Further, we performed logistic regression analysis in which adherence to each practice, within each study arm, was modelled as a function of the time point (baseline and endline) after adjusting for clustering of data within each health facility. The time point p value of these models assessed whether the change in adherence to each practice, within each study arm or group was statistically significant after adjusting for clustering of data within each health facility, To assess whether the change in adherence to each practice from baseline to endline differed significantly between the intervention and comparison groups, we performed the difference in differences (DID) analysis. In this analysis, adherence to each practice was modelled (logistic regression) as a function of intervention status (intervention arm and comparison arm), time point (baseline and endline) and the interaction of these two variables–adjusting for clustering of data within each health facility. The interaction term P value of the multivariate models (for each practice) assessed whether a change from baseline to endline differed significantly between the intervention and comparison groups. All model estimates were computed using robust standard errors. P value of less than 0.05 was considered as statistically significant. The analysis was carried out using MS Excel 2016, Stata version 13, and SPSS version 24.

## Ethics

We received ethical approval for this study from the Institutional Review Board (IRB) of the Johns Hopkins Bloomberg School of Public Health in the United States, the Sigma IRB in India, and the ethical committee of the state government of Odisha in India. The study team obtained permission from the respective state governments to conduct this study at the selected public health facilities.

Data collectors obtained written informed consent from all potential participants who agreed to participate. For illiterate women who could not read the form and sign, data

**Table 1. Practices and steps observed across the four pause points.**

| S no. | Practices | Pause point |
|---|---|---|
| 1 | Client's hemoglobin examination is conducted and the findings are documented. | Pause point 1 (At admission) |
| 2 | Client's medical status is assessed. | |
| 3 | Client's ability to understand the procedure and its consequences is assessed. | |
| 4 | It is confirmed that the client has been fasting for at least six hours. | |
| 5 | Client's pulse, blood pressure, and weight are measured and documented. | |
| 6 | Client's urine examination (for sugar and albumin) is conducted and the findings are documented. | |
| 7 | Client's abdominal and pelvic examinations are conducted. | |
| 8 | Provider interacts directly with client and treats her respectfully | |
| 9 | Provider briefly explains the procedure to the client and encourages her to ask questions | |
| 10 | Provider reads out and explains the consent form to the client in her language | |
| 11 | Client re-confirms her decision to opt for sterilization | |
| 12 | Provider ensures consent form is signed or thumb print is given by the client | |
| 13 | Operating theater (OT) staff changes into OT attire and surgical team performs surgical scrub | Pause point 2 (Pre-operative assessment) |
| | For this practice to be followed, the following steps need to be performed - | |
| | On entering the OT, the OT staff– | |
| | 1. Changes into OT clothes<br>2. Wears OT slippers/shoes<br>3. Wears a cap<br>4. Wears a mask<br>5. The surgeon and the assistants perform surgical scrub as per norms and change into sterile gown before beginning the procedure. | |
| 14 | Surgeon uses sterile gloves and sterile instruments | |
| 15 | Surgeon ensures client has emptied her bladder just before beginning the procedure | |
| 16 | Surgeon provides sedation, analgesia, and local anesthesia as per recommendation. | |
| | For this practice to be followed, the following steps need to be performed– | |
| | 1. Surgeon provides sedation and analgesia using Inj. Fortwin and Phenargan, if not available, gives other appropriate drug/s.<br>2. For local anaesthesia, 2% plain xylocaine is used after diluting with equal amounts of Normal Saline or Distilled Water (to make 1%). | |
| 17 | Incision site is scrubbed adequately | Pause point 3 (Surgery) |
| | For this practice to be followed, the following steps need to be performed - | |
| | 1. Antiseptic solution is applied twice to the incision area.<br>2. Abdomen was cleaned in a circular motion moving outwards from incision area.<br>3. In case of interval ligation, cleaned upper part of pubis and thighs as well.<br>4. In case of postpartum ligation, cleaned the umbilicus first with an antiseptic soaked swab. | |
| 18 | Sterile drapes are used during the surgery | |
| 19 | Surgeon checks for satisfactory anesthetic effect before making incision | |
| 20 | Client's blood pressure and pulse are monitored at least once during surgery | |
| 21 | Client's blood pressure and pulse are documented | |
| 22 | Surgeon follows the recommended surgical technique. | |
| | For this practice to be followed, the following steps need to be performed - | |
| | 1. Both fallopian tubes are identified by tracing up to the fimbrial end.<br>2. Isthmic portion of both fallopian tubes identified, transfixed and cut.<br>3. Catgut is used for ligation. | |

(*Continued*)

**Table 1.** (Continued)

| S no. | Practices | Pause point |
|---|---|---|
| 23 | Client is shifted from OT on a trolley or wheelchair | Pause point 4 (Post-operative care & discharge) |
| 24 | Blood pressure and pulse are monitored post-surgery | |
| 25 | Blood pressure and pulse are documented post-surgery | |
| 26 | Surgical dressing is checked for soakage | |
| 27 | Client is explained about first follow up within 48 hours of surgery | |
| 28 | Client is explained about the second follow up on 7th day after surgery | |
| 29 | Client is explained about the third follow up after one month or next menstrual period | |
| 30 | A filled discharge slip or client card with written instructions is given to the client at discharge | |

collectors signed a witness line on the consent form to confirm that the woman fully understood the study prior to agreeing for participating. In all cases, data collectors made certain that participants understood the contents of the consent form.

## Results

Of the 12 health facilities in the intervention arm, nine were located in Odisha and three in Chhattisgarh, reflecting the greater number of MCSP-supported facilities in Odisha overall. Two of the 12 were district hospitals, one was a sub-district level hospital, and nine were community health centers. The comparison facilities had the same geographic breakdown and included one district hospital and 11 community health centers.

At baseline, 240 observations were completed for each pause point in both intervention and comparison facilities (Table 2). At endline, 240 observations were completed for each pause point in the intervention facilities. However, fewer endline observations were completed at comparison facilities: 223 observations for pause points 1 and 4, and 214 observations for pause points 2 and 3. The shortfall in the planned sample size was due to personnel changes at two health facilities in the comparison arm: the surgeons who performed female sterilizations at these facilities were transferred and not immediately replaced.

Adherence levels for practices observed during pause point 1 (admission) varied widely at baseline. Two practices–urine tests for sugar and albumin and obtaining women's signatures on consent forms–were nearly universal. At the other extreme, adherence was exceptionally low for abdominal and pelvic examinations in both study arms and remained low at endline (25.4% in the intervention group and 16.6% in the comparison group) (Table 3).

Findings suggest the intervention led to improved adherence among six of the 12 practices observed during pause point 1. This included three practices related to women's medical eligibility for female sterilization, two practices related to client-provider interaction, and one practice

**Table 2. Number of completed observations at baseline and endline, by study arm and pause point.**

| Round | Study arm | Pause point | | | |
|---|---|---|---|---|---|
| | | 1 | 2 | 3 | 4 |
| Baseline | Intervention | 240 | 240 | 240 | 240 |
| | Comparison | 240 | 240 | 240 | 240 |
| Endline | Intervention | 240 | 240 | 240 | 240 |
| | Comparison | 223 | 214 | 214 | 223 |

**Table 3. Percent of observations at pause point 1 (admission) in which providers adhered to key practices, by study arm and round of data collection (n = 943).**

| Practices and study arm | Bivariate analysis | | | | Multivariate analysis[b] | |
|---|---|---|---|---|---|---|
| | % achieved | | Change from baseline to endline | | Adjusted p-value for change within group | p-value for interaction |
| | Baseline | Endline | % points | p-value[a] | | |
| **Ascertaining client's medical eligibility for undergoing sterilization:** | | | | | | |
| **Client's hemoglobin examination done and documented** | | | | | | |
| Comparison | 96.3 | 97.3 | +1 | 0.605 | 0.663 | **0.028** |
| Intervention | 89.6 | 99.6 | +10 | **0.001** | **0.003** | |
| **Assessment of client's medical status** | | | | | | |
| Comparison | 72.5 | 46.6 | -25.9 | **0.001** | 0.162 | **0.003** |
| Intervention | 55.8 | 92.1 | +36.3 | **0.001** | **0.006** | |
| **Assessment of client's ability to understand the procedure and its consequences** | | | | | | |
| Comparison | 47.9 | 26.9 | -21 | **0.001** | 0.269 | **0.015** |
| Intervention | 39.2 | 79.7 | +40.5 | **0.001** | **0.018** | |
| **Confirmation that the client has been fasting for at least six hours** | | | | | | |
| Comparison | 35.8 | 30.9 | -4.9 | 0.279 | 0.770 | 0.243 |
| Intervention | 52.9 | 74.7 | +21.8 | **0.001** | 0.166 | |
| **Client's pulse, blood pressure, and weight measured and documented** | | | | | | |
| Comparison | 12.1 | 27.8 | +15.7 | **0.001** | 0.267 | 0.622 |
| Intervention | 24.2 | 60.6 | +36.4 | **0.001** | **0.011** | |
| **Client's urine examination (for sugar and albumin) done and documented** | | | | | | |
| Comparison | 97.9 | 96.9 | -1 | 0.565 | 0.640 | 0.197 |
| Intervention | 97.1 | 99.2 | +2.1 | 0.106 | 0.192 | |
| **Abdominal and pelvic examination of client** | | | | | | |
| Comparison | 8.8 | 10.3 | +1.5 | 0.635 | 0.893 | 0.660 |
| Intervention | 25.4 | 16.6 | -8.8 | **0.019** | 0.571 | |
| **Client provider interaction:** | | | | | | |
| **Provider interacts directly with client and treats her respectfully** | | | | | | |
| Comparison | 75.0 | 17.5 | -57.5 | **0.001** | **0.001** | **0.002** |
| Intervention | 52.1 | 61.4 | +9.3 | **0.043** | 0.542 | |
| **Provider briefly explains the procedure to the client and encourages her to ask questions** | | | | | | |
| Comparison | 30.4 | 9 | -21.4 | **0.001** | **0.037** | **0.018** |
| Intervention | 22.1 | 40.7 | +18.6 | **0.001** | 0.224 | |
| **Consent process:** | | | | | | |
| **Consent form is read out and explained to the client in her language** | | | | | | |
| Comparison | 9.2 | 1.8 | -7.4 | **0.001** | 0.055 | **0.023** |
| Intervention | 23.3 | 41.1 | +17.8 | **0.001** | 0.234 | |
| **Client re-confirms her decision to opt for sterilization** | | | | | | |
| Comparison | 28.7 | 27.8 | -0.9 | 0.837 | 0.957 | 0.208 |
| Intervention | 34.2 | 66.4 | +32.2 | **0.001** | 0.056 | |
| **Provider ensures consent form is signed or thumb print is given by the client** | | | | | | |
| Comparison | 99.2 | 97.8 | -1.4 | 0.270 | 0.187 | 0.225 |
| Intervention | 90.4 | 95.9 | +5.5 | **0.020** | 0.518 | |

Note: 943 observations include 240 observations in intervention facilities at baseline, 240 observations in intervention facilities at endline, 240 observations in comparison facilities at baseline, and 223 observations in comparison facilities at endline.

[a] Chi square test

[b] Adjusted for clustering by health facilities

related to obtaining informed consent from clients. For five of these practices, the bivariate analysis found that adherence declined significantly at comparison facilities while increasing significantly at intervention facilities. Despite positive changes, considerable room for improvement remained for some practices. For example, providers explained the procedure to the client and encouraged questions in just 40.7% of endline sessions at intervention facilities.

Adherence levels also varied widely at pause points 2 and 3 (pre-operative assessment and surgery). Baseline adherence levels were below 30% for sedation and anesthesia and also for monitoring and documenting vital signs during surgery (Table 4). In contrast, surgeons used

**Table 4. Percent of observations at pause points 2 and 3 (pre-operative assessment and surgery) in which providers adhered to key practices, by study arm and round of data collection (n = 934).**

| Practices and study arm | Bivariate analysis | | | | Multivariate analysis[b] | |
|---|---|---|---|---|---|---|
| | % achieved | | Change from baseline to endline | | Adjusted p-value for change within group | p-value for interaction |
| | Baseline | Endline | % points | p-value[a] | | |
| **Operating theater (OT) staff changes into OT attire and surgical team performs surgical scrub** | | | | | | |
| Comparison | 57.5 | 47.2 | -10.3 | **0.031** | 0.600 | |
| Intervention | 63.7 | 44.8 | -18.9 | **0.001** | 0.295 | 0.738 |
| **Surgeon uses sterile gloves and sterile instruments** | | | | | | |
| Comparison | 86.3 | 70.1 | -16.2 | **0.001** | 0.164 | 0.822 |
| Intervention | 90.0 | 80.5 | -9.5 | **0.004** | 0.187 | |
| **Surgeon ensures client has emptied her bladder just before beginning the procedure** | | | | | | |
| Comparison | 81.7 | 74.8 | -6.9 | 0.087 | 0.571 | 0.766 |
| Intervention | 88.3 | 78.4 | -9.9 | **0.005** | 0.383 | |
| **Surgeon provides sedation, analgesia, and local anesthesia as per recommendation** | | | | | | |
| Comparison | 2.1 | 15.9 | +13.8 | **0.001** | **0.011** | 0.454 |
| Intervention | 26.7 | 58.5 | +31.8 | **0.001** | 0.057 | |
| **Incision site was scrubbed adequately** | | | | | | |
| Comparison | 43.8 | 9.8 | -34 | **0.001** | **0.002** | 0.076 |
| Intervention | 44.2 | 36.9 | -7.3 | 0.115 | 0.666 | |
| **Sterile drapes were used** | | | | | | |
| Comparison | 93.3 | 71.5 | -21.8 | **0.001** | 0.110 | 0.090 |
| Intervention | 93.8 | 97.1 | +3.3 | 0.085 | 0.444 | |
| **Surgeon checks for satisfactory anesthetic effect before making incision** | | | | | | |
| Comparison | 35.4 | 54.7 | +19.3 | **0.001** | 0.267 | 0.129 |
| Intervention | 63.3 | 48.5 | -14.8 | **0.001** | 0.312 | |
| **Client's blood pressure and pulse are monitored at least once during surgery** | | | | | | |
| Comparison | 0.4 | 0 | -0.4 | 1.000 | 0.934 | 0.867 |
| Intervention | 22.5 | 19.9 | -2.6 | 0.505 | 0.859 | |
| **Client's blood pressure and pulse are documented** | | | | | | |
| Comparison | 0.4 | 0 | -0.4 | 1.000 | 0.934 | 0.772 |
| Intervention | 15.4 | 24.9 | +9.5 | **0.012** | 0.532 | |
| **Surgeon follows recommended surgical technique** | | | | | | |
| Comparison | 85.8 | 90.7 | +4.9 | 0.146 | 0.721 | 0.722 |
| Intervention | 97.5 | 99.2 | +1.7 | 0.176 | 0.387 | |

Note: 943 observations include 240 observations in intervention facilities at baseline, 240 observations in intervention facilities at endline, 240 observations in comparison facilities at baseline, and 223 observations in comparison facilities at endline.

[a] Chi square test

[b] Adjusted for clustering by health facilities

sterile gloves, instruments, and drapes, ensured the client had emptied her bladder, and followed recommended surgical technique in over 80% of cases at baseline, limiting the room for improvement. A comparison of the two study arms from baseline to endline shows no significant impact of the intervention on any practices during pause points 2 and 3. The bivariate analysis found that adherence actually fell significantly at intervention sites for four practices.

At baseline, adherence to all key practices observed during pause point 4 (post-operative care and discharge) was low, ranging from 0 to 45%. In the intervention group, the bivariate analysis showed significant increases in adherence to all practices, while adherence in the comparison group remained static or declined. Changes in three practices in the intervention group were significant in the multivariate analysis: documenting the client's vitals, checking the surgical dressing, and giving clients a discharge slip or card with written instructions. At endline, however, adherence levels in the intervention group exceeded 50% for only two practices: use of a trolley or wheelchair (63.5%) and discharge slips (86.3%) (Table 5).

**Table 5. Percent of observations at pause point 4 (post-operative care and discharge) in which providers adhered to key practices, by study arm and data collection round (n = 943).**

| Practices and study arm | Bivariate analysis | | | | Multivariate analysis** | |
|---|---|---|---|---|---|---|
| | % achieved | | Change from baseline to endline | | Adjusted p-value for change within group | p-value for interaction |
| | Baseline | Endline | % points | p-value* | | |
| **Client shifted from operating theater on a trolley or wheelchair** | | | | | | |
| Comparison | 28.3 | 18.4 | -9.9 | **0.012** | 0.556 | |
| Intervention | 45.4 | 63.5 | +18.1 | **0.001** | 0.334 | 0.283 |
| **Blood pressure and pulse monitored post-surgery** | | | | | | |
| Comparison | 0.4 | 0 | -0.4 | 1.000 | 0.958 | 0.234 |
| Intervention | 7.5 | 36.1 | +28.6 | **0.001** | **0.012** | |
| **Blood pressure and pulse documented post-surgery** | | | | | | |
| Comparison | 0.4 | 0 | -0.4 | 1.000 | 0.993 | **0.011** |
| Intervention | 0.8 | 34 | +33.2 | **0.001** | **0.001** | |
| **Surgical dressing checked for soakage** | | | | | | |
| Comparison | 22.5 | 0 | -22.5 | **0.001** | **0.001** | |
| Intervention | 9.2 | 18.7 | +9.5 | **0.004** | 0.302 | **0.001** |
| **Explained to client about first follow up within 48 hours of surgery** | | | | | | |
| Comparison | 10.0 | 4 | -6 | **0.018** | 0.447 | 0.155 |
| Intervention | 21.7 | 46.1 | +24.4 | **0.001** | 0.142 | |
| **Explained to client about second follow up on 7th day after surgery** | | | | | | |
| Comparison | 6.7 | 4 | -2.7 | 0.225 | 0.716 | 0.278 |
| Intervention | 22.1 | 49 | +26.9 | **0.001** | 0.096 | |
| **Explained to client about third follow up after one month or next menstrual period** | | | | | | |
| Comparison | 6.7 | 4 | -2.7 | 0.225 | 0.716 | 0.140 |
| Intervention | 11.7 | 47.3 | +35.6 | **0.001** | **0.021** | |
| **Filled discharge slip or client card with written instructions and gave to client** | | | | | | |
| Comparison | 19.6 | 9 | -10.6 | **0.001** | 0.457 | **0.017** |
| Intervention | 31.7 | 86.3 | +54.6 | **0.001** | **0.003** | |

Note: 943 observations include 240 observations in intervention facilities at baseline, 240 observations in intervention facilities at endline, 240 observations in comparison facilities at baseline, and 223 observations in comparison facilities at endline.

*—by Chi square test,

**—after adjusting for clustering by health facilities

## Discussion

This quasi-experimental study examined the effectiveness of a package of interventions–supported by MCSP and implemented in collaboration with GoI–in improving providers' adherence to key evidence-based practices during female sterilization services offered at public health facilities in two Indian states. We found that a few of the 30 practices observed were well established prior to the intervention, including hemoglobin and urine testing, using sterile gloves and surgical instruments, and following recommended surgical techniques. However, adherence to many other practices was extremely low at baseline. The package of program-supported interventions (which included a competency-based hands-on training, mentoring, introduction and use of a safety checklist, and a client card) improved adherence to nine practices, including practices related to ascertaining client's medical eligibility, client-provider interaction, the consent process, and post-surgery care. The greatest improvement was observed in giving women written instructions prior to discharge.

Ascertaining the medical eligibility of women prior to undergoing sterilization surgery requires both laboratory and clinical screening practices. Hemoglobin and urine examinations prior to surgery were completed in most observed cases even at baseline, which is consistent with earlier studies of female sterilization surgeries in public health facilities in India [13, 27, 28]. The intervention also led to more consistent assessments of the client's medical status. However, it is equally important to ensure that clients comprehend the implications of their decision because female sterilization permanently limits future childbearing. This is central to an informed and voluntary approach to choosing this or (any) other family planning methods. Our study found that less than half of the clients at baseline were assessed for their ability to understand the consequences of accepting a permanent family planning method. This corroborates with the findings of a recent study from India [27], which also reported low adherence to this critical practice. The MCSP intervention led to substantial gains, as providers at intervention facilities assessed the comprehension of four-fifths of women at the endline.

Abdominal and pelvic examination findings provide valuable information on the presence of adhesions or adnexal masses that portend a difficult surgery. At baseline, these exams were conducted in just one-quarter or fewer of observed cases, and the intervention did not improve adherence to this practice. This may have been due to various implementation challenges, including a lack of doctors to perform these examinations, lack of competency to correctly perform and interpret meaningful findings, and lack of private space at facilities' screening sites. While other steps in screening women for female sterilization can be performed by nursing or paramedical staff (including doctors of the Indian system of medicine), abdominal and pelvic examinations require a medical doctor as per GoI guidelines. It should be noted that routine screening of non-pregnant, asymptomatic women with pelvic exams poses implementation challenges even in high-income countries [29]. From the woman's perspective, these intimate physical examinations hold the potential for embarrassment, anxiety, and discomfort. Doctors also have anxieties, including a lack of confidence in their clinical findings and fear of alleged misconduct [29].

Respectful care and good client-provider interaction are critical to a woman's initial decision to adopt a family planning method and later, to adhere to the chosen method of contraception [30]. Effective communication, respect and dignity, and emotional support comprise the three domains of client experience of care in the WHO Quality of Care framework [31]. This study directly measured two critical elements of respectful care: providers interacting directly and respectfully with women and providing them with full explanations of the procedure. Baseline scores for interacting directly with clients and treating them respectfully were surprisingly high (52.1% in the intervention arm and 75% in comparison arm) compared with

previous studies in India [4, 13]. This may be due, in part, to providers being aware of the presence of observers and modifying their behaviors accordingly (the Hawthorne effect) [32]. At endline, adherence to this practice dropped significantly in the comparison arm while it improved further in the intervention arm. Similar trends were seen for the practice of explaining the procedure to the client and encouraging her to ask questions, although adherence levels were lower.

Obtaining written consent from women prior to surgery was almost universal, which is similar to findings reported by other studies in India [13, 27, 28, 33]. A likely explanation is that a signed consent form is part of the documentation required for reimbursement of clients and providers. However, studies have reported that little effort goes into ensuring that women actually understand the contents of the consent form and thus make a truly informed decision [13, 28]. Our baseline findings corroborated this concern: less than one-fourth of observed women received an explanation of the contents of the consent form. The practice of explaining the consent form–which includes providing information on other available highly effective long acting reversible methods, the potential for failure, permanence of the procedure, risks of surgery, the lack of protection against sexually transmitted infections, and the choice to opt out of the procedure without losing access to other medical treatment–is central to a client-centered, quality approach to providing family planning and reproductive health care [20]. In India, a national scheme provides a fixed payment to both the client and the field-level worker who accompanies her; it is designed to offset the loss of wages and other out-of-pocket expenses that may be incurred due to sterilization. In this context, it is imperative that a client's decision to undergo this surgery is well informed and voluntary, free from coercion or the lure of financial incentives. Although the intervention did lead to a significant improvement in explaining the consent form, adherence remained less than 50% at endline, clearly pointing to the need for further improvement. Inadequate staff, even at intervention facilities, posed a major barrier to this process. Fewer staff catering to higher caseloads on FDS days leaves little opportunity for concerned staff to explain the contents of the consent form to every client individually.

The intervention did not have an effect on any of the key practices observed during the pre-operative assessment and surgery (pause points 2 and 3). Adherence to a few of these practices was high at baseline, but adherence to other practices was low at baseline and changed little. For certain practices, such as monitoring and documenting vital signs during surgery, a lack of adequately trained staff in the operating theater may have posed a barrier to adherence. Using pulse oximeters for monitoring vital signs, which is an established standard in the WHO Safe Surgery Checklist, could facilitate adherence to this key practice [34].

Practices around infection prevention and control did not change after the intervention. At endline, for example, changing into proper attire and performing a surgical scrub before entering the operating theater was observed in less than half of cases, while adequate scrubbing of the incision site was observed in less than one-fourth of cases. Literature from around the world points out that improving adherence to infection prevention practices requires modifying entrenched behaviors of healthcare workers and has always been a big challenge [35, 36]. Experts in the field advocate for multimodal approaches that utilize behavior change models, some of which have been effective in modifying behaviors of health service providers in different clinical situations [36]. Our intervention–which focused on imparting knowledge and skills on infection prevention control through the trainings and mentoring–could have benefitted from the inclusion of behavior change strategies.

Consistent improvements were observed in post-surgical care, although not all gains proved significant in the multivariate analysis. Periodic and appropriate monitoring after sterilization is imperative for the timely detection of anesthesia- and/or procedure-related complications.

Prior studies in India corroborate our baseline findings showing poor adherence to post-surgery monitoring at public health facilities [4, 13, 27]. The intervention led to significant gains in practices like checking the surgical dressing for soakage and documenting the vital signs of clients before discharge. Despite improvements, however, adherence remained low at endline (less than one-third). Earlier studies have pointed to a lack of nursing staff as the reason for poor adherence to post-operative monitoring [13], and staff shortages may also have played a role in this study. Government reports based on healthcare administrative data point to a lack of nursing and paramedical staff at FDS sites (including community health centers, sub-district/divisional hospitals, and district hospital) in these two states [37].

Notably, the intervention almost tripled the proportion of female sterilization clients who received written instructions prior to their discharge from the facility. Earlier studies in India reported poor adherence to this practice [4, 13], which were confirmed by our baseline findings showing that less than one-third of women were given written instructions prior to discharge. By the endline, that proportion had climbed to 86%. This can be attributed to the introduction of a printed client card at intervention facilities, along with increased emphasis on the importance of giving these instructions during the MCSP program.

To the best of our knowledge, this study is first of its kind from India. A major strength is its use of direct observations of clinical care by senior surgeons and experienced nurses or ANMs. This is considered the gold standard for measuring quality of care [38]. Previous published studies of the quality of female sterilization services in India have relied on record audits or client interviews which are liable to biases.

Both the program and the study were conducted in close coordination with the state health departments, and there were no other quality improvement interventions with similar objectives implemented at the study sites, during course of the study.

However, our study does have certain limitations. Direct observations of service providers may have biased some findings due to the Hawthorne effect, though the observers were trained to make their observations in an as unobtrusive manner as possible Inter-observer variation may have also had an influence on the findings, although rigorous procedures for training data collectors sought to avoid this, including standardized trainings by the same instructors before each round of data collection. In addition, during the course of this study, we were able to retain more than 70% of data collectors from baseline to endline. However, inter-rater reliability was not measured as part of the study. Lastly, three comparison facilities had to be replaced in the endline round of data collection after they stopped performing the minilap procedure, although efforts were made to ensure that the additional facilities were comparable to the others, based on the same matching criteria and there were no quality improvement interventions, with similar objectives as MCSP, being implemented there. There was also a shortfall in the number of observations completed in comparison facilities at endline (874 observations were conducted of the 960 originally planned), but the missing observations were spread relatively evenly across the four pause points. Given these limitations, a degree of caution should be exercised in interpreting the data.

## Conclusion

This study highlights the need for quality improvement interventions like the one implemented by MCSP, as adherence to many key practices identified in the Government of India's standards and guidelines for female sterilization was very low at baseline. Although the package of interventions studied here did improve adherence to certain key practices around admission, post-operative care and discharge, many practices–even those that saw an improvement–remained low at endline. Contextual factors such as insufficient human resources,

frequent turn-over of trained staff, and inadequate infrastructure may have blunted the potential impact of the intervention.

Utilization of behavior change models as part of the intervention might have helped improve practices around infection prevention and control. In addition, inclusion of respectful care as an essential aspect of family planning service provision and measures to ensure reduced turn-over of trained staff can further strengthen adherence to best-practices and ensure sustainability of quality improvement initiatives such as MCSP.

Further research is needed to understand factors that can enable more consistent adherence to evidence-based practices in such resource-constrained health facilities.

## Supporting information

**S1 Checklist. Assessment checklist for quality of family planning study.**
(DOCX)

**S1 Dataset.**
(XLSX)

## Acknowledgments

The authors would like to thank India's Ministry of Health and Family Welfare and USAID for their invaluable support and guidance and to acknowledge Academy of Management Studies, Lucknow, India, which was responsible for data collection. The authors are grateful to the government health officials of the states of Chhattisgarh and Odisha for their support during the study. The authors also extend their sincere thanks to all the people who participated in the study. The authors would also like to thank Adrienne Kols for critically reviewing the manuscript and providing valuable inputs.

## Author Contributions

**Conceptualization:** Ashish Srivastava, Geeta Chhibber, Neeta Bhatnagar, Vivek Yadav, Bulbul Sood.

**Data curation:** Ashish Srivastava.

**Formal analysis:** Ashish Srivastava.

**Investigation:** Ashish Srivastava, Geeta Chhibber.

**Methodology:** Ashish Srivastava, Barbara Rawlins.

**Project administration:** Geeta Chhibber, Angela Nash-Mercado, Jyoti Samal, Bhagyashree Trivedi, Vivek Yadav.

**Supervision:** Ashish Srivastava, Geeta Chhibber, Jyoti Samal, Bhagyashree Trivedi, Vinod Srivastava.

**Validation:** Ashish Srivastava.

**Writing – original draft:** Ashish Srivastava, Geeta Chhibber, Neeta Bhatnagar, Angela Nash-Mercado, Jyoti Samal, Bhagyashree Trivedi, Vinod Srivastava, Vivek Yadav.

**Writing – review & editing:** Ashish Srivastava, Geeta Chhibber, Neeta Bhatnagar, Angela Nash-Mercado, Barbara Rawlins, Regien Biesma, Young-Mi Kim, Jelle Stekelenburg.

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
