## [Decision Letter · Decision Letter 0]

16 Oct 2020

PONE-D-20-30411

Effectiveness of a quality improvement intervention to increase adherence to key practices during female sterilization services in Chhattisgarh and Odisha states of India

PLOS ONE

Dear Dr. Chhibber,

Thank you for submitting your manuscript to PLOS ONE. After careful consideration, we feel that it has merit but does not fully meet PLOS ONE’s publication criteria as it currently stands. Therefore, we invite you to submit a revised version of the manuscript that addresses the points raised during the review process.

We look forward to receiving your revised manuscript.

Kind regards,

Vijayaprasad Gopichandran

Academic Editor

PLOS ONE

Journal Requirements:

2. In the Methods, please clarify that participants provided oral consent. Please also state in the Methods:

- Why written consent could not be obtained

- Whether the Institutional Review Board (IRB) approved use of oral consent

- How oral consent was documented

For more information, please see our guidelines for human subjects research: https://journals.plos.org/plosone/s/submission-guidelines#loc-human-subjects-research

4. We note you have included a table to which you do not refer in the text of your manuscript. Please ensure that you refer to Table 4 in your text; if accepted, production will need this reference to link the reader to the Table.

Reviewers' comments:

Reviewer's Responses to Questions

**Comments to the Author**

1. Is the manuscript technically sound, and do the data support the conclusions?

Reviewer #1: Yes

Reviewer #2: Yes

2. Has the statistical analysis been performed appropriately and rigorously? 

Reviewer #1: Yes

Reviewer #2: Yes

3. Have the authors made all data underlying the findings in their manuscript fully available?

Reviewer #1: No

Reviewer #2: Yes

4. Is the manuscript presented in an intelligible fashion and written in standard English?

Reviewer #1: Yes

Reviewer #2: Yes

5. Review Comments to the Author

Reviewer #1: India is a young country with 65 percent of the population below the age of 35 years. Increasing population still remains a challenge for some states, where an increased burden is posed by the unwanted child bearing, which has been extensively highlighted in the existing literature. Contraception is a cost-effective method to curb the growth in population size. Especially, in this era of COVID-19 where the unintended baby boom is a possibility, effective use of contraception becomes a necessity. Sterilization being one of the most effective contraception method should be promoted extensively, however, the major challenge in the domain still remains the poor quality of care in sterilization operation. As study based on NFHS India suggests that sterilization regret due to bad quality of care had increased from 13% in NFHS-1 to 16% in NFHS-4. In view of this the present study is extremely important. However, there are few observations which will improve the readability of the manuscript. The observations are as follows:

1. What "know-do-gap" are authors referring to. Give a detailed description of the same.

2. What criteria were followed to screen the fitness of the women (Line 136).

3. Give a detailed description of sample size calculation along with the formula used.

4. The sample selection (although explained in the text), would be benefitted if presented in a flow chart or figure form.

5. Authors should revise the Data Method section completely. Add the variable heading and the checklist in the main text if possible.

6. The section on statistical analysis need to be more elaborate and reorganized.

7. The study is based on a quasi-experimental design, which is generally, an experimental design missing one or more of its characteristics. It would be better to explain why are the authors referring this to as a quasi-experimental design? Is it because there was no random assignment?

8. The participants in both the arms were matched, what were the characteristics used to match these participant? Please provide an elaborate explanation.

9. Add few lines on policy implications. Are there any suggestions which can help in improving the quality of sterilization care?

10. The authors mention the Ethics, and consent taken form the study participant in two separate places in the manuscript. It would be better to mention these information as one single paragraph under Ethics.

11. What were the strengths and limitations of the study?

12. There are minor error in the reference style. Please check and rectify.

13. There are minor English language issues. The manuscript can be benefitted with proper proof-reading.

14. The study importance, findings, conceptualization and statistical analysis are adequate to address the study objectives.

Reviewer #2: Dear authors,

Overall, this was a good research work, and it highlights an important issue of the quality of care of sterilization services in India. This paper provides insights into what is happening at the bottom level of family planning service delivery. I have some suggestions and hope it will further improve your paper.

1. As this study aims to examine whether the MCS program interventions increased adherence to female sterilization services at public health facilities in two Indian states, it would be of use to add a brief note in the introduction section on what is the MCS program and how it is implemented in the country.

2. It is always challenging to evaluate a particular intervention and program as multiple programs and interventions with similar aims and objectives runs simultaneously. How did you ensure whether no interventions with similar objectives under NRHM, state health departments or any other organization overlap with the goals of this study?

3. Many of the existing literature and researches suggest better training to service providers and maintaining safety checklist ensures good quality service delivery in family planning services. This study showed that providing better training and maintaining safety checklist not always guarantee good service delivery, especially in the case of pre-operative assessment and surgery. In this context, authors could specifically suggest what could be done further to improve the quality of sterilization services in the country.

4. Quasi-experimental designs are subject to a variety of selection related threats such as:

• Selection-history threat (the intervention and comparison groups being differentially impacted by extraneous or historical events),

• Selection-regression threat (the intervention and comparison groups regressing toward the mean between baseline and end-line at different rates)

• Selection-instrumentation threat (the intervention and comparison groups responding differently to the measurement)

• Selection-testing (the intervention and comparison groups responding differently to the baseline)

• Selection-mortality (the intervention and comparison groups demonstrating differential dropout rates. In this study there was a shortfall in the number of observations completed in comparison facilities at end-line)

What measures were adopted in this study to overcome those threats mentioned above?

6. PLOS authors have the option to publish the peer review history of their article (what does this mean?). If published, this will include your full peer review and any attached files.

Reviewer #1: No

Reviewer #2: No

---

## [Author Response · Author response to Decision Letter 0]

2 Dec 2020

Comment no. Reviewer Response Line no.

(In the track change version)

We would like to thank the editor and reviewers for their valuable comments and suggestions. They have helped us in improving our manuscript further.

Below are our comment wise responses to the comments and suggestions.

Comments from the editor

1 Please ensure that your manuscript meets PLOS ONE's style requirements, including those for file naming. Many thanks for your feedback. We have made changes in order to meet PLOS ONEs requirements. NA

2 In the Methods, please clarify that participants provided oral consent. Please also state in the Methods:

• Why written consent could not be obtained. You may want also to include

• Whether the Institutional Review Board (IRB) approved use of oral consent

• How oral consent was documented

 Many thanks for this important point. We did obtain written consent from all participants except illiterate women, who could not read and sign the document. For such women, the data collectors explained the contents of the form in detail and once they agreed to participate, signed a witness line on the consent form to confirm that the woman fully understood the study prior to agreeing to participate in the study. This was as per the suggestion of the local IRB in India. 

We have made changes to add more clarity on this.

The added changes are as follows – “Data collectors obtained written informed consent from all potential participants who agreed to participate. For illiterate women who could not read the form and sign, data collectors signed a witness line on the consent form to confirm that the woman fully understood the study prior to agreeing for participating. In all cases, data collectors made certain that participants women understood the contents of the consent form” 284 – 288

3 PLOS requires an ORCID iD for the corresponding author in Editorial Manager on papers submitted after December 6th, 2016. Please ensure that you have an ORCID iD and that it is validated in Editorial Manager. As suggested, the corresponding author has now registered on ORCID. 

The id of corresponding author is – 0000-0003-2526-2405 

 NA

4 We note you have included a table to which you do not refer in the text of your manuscript. Please ensure that you refer to Table 4 in your text; if accepted, production will need this reference to link the reader to the Table.

 We have added the reference to table 4 (now table 5) in the text. 367

Comments from Reviewer 1

1 What "know-do-gap" are authors referring to? Give a detailed description of the same.

 Many thanks for the comments. We have elaborated on the “know-do-gap” and also added references for the same. 

Added section – “While increased knowledge of health service providers is often seen as a process indicator of improved quality of care, it may not necessarily translate into change in practices in the real-world settings [22, refs]. Providers may not ‘do’ as per what they ‘know’. Therefore understanding the know-do gap is a critical step towards developing effective, practical strategies to improve delivery of quality female sterilization services.”

 106 – 111

2 What criteria were followed to screen the fitness of the women (Line 136).

 The criteria included blood pressure measurement, hemoglobin estimation, urine examination for proteins and sugar and ruling out pregnancy using the pregnancy checklist. We have added these to the manuscript.

Added section – “Then community-based health workers (Auxiliary Nurse Midwives or ANMs) used the card to screen women’s fitness for the sterilization procedure that included a blood pressure measurement, hemoglobin estimation, urine for proteins and sugar and ruling out pregnancy using the pregnancy checklist.” 150-151

3 Give a detailed description of sample size calculation along with the formula used.

 We have added more details on the sample size calculation. We have also included the formula which was used for sample size estimation.

Added and revised section – “For estimating the sample size, we used the following formula N = 2 (Zα/2 + Zβ) 2 P (1 – P) / (p1 – p2)2, which is for comparison between two groups when the endpoint is qualitative (ref). When calculating the sample size, we assumed that health service providers adhered to 50% of key female sterilization practices at baseline (p1), using 80% power (1 – β) to detect a 20% change (p1 – p2) in adherence at endline with 5% type I error (α). To account for clustering by health facilities, we inflated the sample size using a design effect of 2.51.19 (cluster size of 20, intra cluster correlation coefficient of 0.081). We rounded up the calculated sample size of 23523 observations to 240 observations in order to evenly distribute them among 12 clusters (i.e., facilities) for each study group in each round. This resulted in a total sample size of 960 observations: 240 observations per study arm per round of data collection. These were evenly divided across facilities, so that 20 observations were made at each facility at baseline and again at endline.”

 187 – 194

4 The sample selection (although explained in the text), would be benefitted if presented in a flow chart or figure form.

 Thank you for this suggestion. We have added a flow chart for more clarity. Figure 1

5 Authors should revise the Data Method section completely. Add the variable heading and the checklist in the main text if possible.

 In line with your suggestions, we have added a separate table (Table 1) to the section detailing on the practices and steps observed. As suggested, we have also added the observation checklist as a supplementary file.

 Table 1 & supplementary file 1

6 The section on statistical analysis need to be more elaborate and reorganized.

 We agree with this comment and have added some more details and reorganized the section for more clarity.

Added section – “We computed the proportion of observations in which the provider adhered to a key practice and compared proportions at endline and baseline in each study arm by performing Chi square tests. Further, we performed logistic regression analysis in which adherence to each practice, within each study arm, was modelled as a function of the time point (baseline and endline) after adjusting for clustering of data within each health facility. The time point p value of these models assessed whether the change in adherence to each practice, within each study arm or group was statistically significant after adjusting for clustering of data within each health facility, To assess whether the change in adherence to each practice from baseline to endline differed significantly between the intervention and comparison groups, we performed the difference in differences (DID) analysis. In this analysis, adherence to each practice was modelled (logistic regression) as a function of intervention status (intervention arm and comparison arm), time point (baseline and endline) and the interaction of these two variables – adjusting for clustering of data within each health facility. The interaction term P value of the multivariate models (for each practice) assessed whether a change from baseline to endline differed significantly between the intervention and comparison groups. All model estimates were computed using robust standard errors. P value of less than 0.05 was considered as statistically significant. The analysis was carried out using MS Excel 2016, Stata version 13, and SPSS version 24.”

 254 - 276

7 The study is based on a quasi-experimental design, which is generally, an experimental design missing one or more of its characteristics. It would be better to explain why are the authors referring this to as a quasi-experimental design? Is it because there was no random assignment?

 Thank you for the question. Yes, it is primarily because there was no random assignment. The intervention districts and facilities were identified in consultation with the respective state governments based on their priorities and performance of districts on some key FP indicators. 

We have now specifically added this information for more clarity. 

Added text – “It had a quasi-experimental design as the intervention districts for MCSP were selected purposefully and not randomly. The intervention districts were selected in consultation with state officials based on government priorities and poor facility performance on key family planning indicators.” 160 – 163

8 The participants in both the arms were matched, what were the characteristics used to match these participant? Please provide an elaborate explanation.

 Many thanks for your comments. We matched the health facilities a priori. The criteria used for matching the facilities were facility’s (1) state, (2) delivery volume, and (3) average monthly female sterilization client load during the three months prior to baseline data collection. 

Relevant text: “To form a comparison group, we matched the intervention facilities with 12 FDS facilities from nine districts that were not supported by MCSP, based on the facility’s (1) state, (2) delivery volume, and (3) average monthly female sterilization client load during the three months prior to baseline data collection.” 179 – 184

9 Add few lines on policy implications. Are there any suggestions which can help in improving the quality of sterilization care?

 We have added a few suggestions in the conclusion section. These include inclusion of respectful care as an essential aspect of family planning service provision, incorporation of behavior change strategies in clinical trainings and reduced turn-over of trained staff.

Added text: “Utilization of behavior change models as part of the intervention might have helped improve practices around infection prevention and control. In addition, inclusion of respectful care as an essential aspect of family planning service provision and measures to ensure reduced turn-over of trained staff can further strengthen adherence to best-practices and ensure sustainability of quality improvement initiatives such as MCSP.” 528 - 532

10 The authors mention the Ethics, and consent taken form the study participant in two separate places in the manuscript. It would be better to mention these information as one single paragraph under Ethics.

 Thank you for the suggestion. We have now included all the ethics related information under one paragraph 279 - 288

11 What were the strengths and limitations of the study?

 We have elaborated on the strengths and limitations of the study in the last paragraph of the discussion section.

Relevant Section: “To the best of our knowledge, this study is first of its kind from India. A major strength is its use of direct observations of clinical care by senior surgeons and experienced nurses or ANMs. This is considered the gold standard for measuring quality of care [35]. Previous published studies of the quality of female sterilization services in India have relied on record audits or client interviews which are liable to biases. Both the program and the study were conducted in close coordination with the state health departments, and there were no other quality improvement interventions with similar objectives implemented at the study sites, during course of the study. 

However, our study does have certain limitations. Direct observations of service providers may have biased some findings due to the Hawthorne effect, though the observers were trained to make their observations in an as unobtrusive manner as possible. Inter-observer variation may have also had an influence on the findings, although rigorous procedures for training data collectors sought to avoid this, including standardized trainings by the same instructors before each round of data collection. In addition, during the course of this study, we were able to retain more than 70% of data collectors from baseline to endline. However, inter-rater reliability was not measured as part of the study. Lastly, three comparison facilities had to be replaced in the endline round of data collection after they stopped performing the minilap procedure, although efforts were made to ensure that the additional facilities were comparable to the others, based on the same matching criteria and there were no quality improvement interventions, with similar objectives as MCSP, being implemented there.. There was also a shortfall in the number of observations completed in comparison facilities at endline (874 observations were conducted of the 960 originally planned), but the missing observations were spread relatively evenly across the four pause points. Given these limitations, caution should be exercised in interpreting the data. “ 494 - 517

12 There are minor error in the reference style. Please check and rectify.

 We have re-checked the references and done the necessary corrections. 

 NA

13 There are minor English language issues. The manuscript can be benefitted with proper proof- reading.

 We have done the proof reading and corrected the language issues to the best of our ability.

 NA

14 The study importance, findings, conceptualization and statistical analysis are adequate to address the study objectives.

 Thank you for the comments and all the valuable inputs which have helped make our paper better. 

 NA

Comments from Reviewer 2 

1 As this study aims to examine whether the MCS program interventions increased adherence to female sterilization services at public health facilities in two Indian states, it would be of use to add a brief note in the introduction section on what is the MCS program and how it is implemented in the country.

 Thank you for the suggestion. We have added a brief note on the MCSP program as suggested.

Added section – “MCSP was a global program focused in 26 high priority-countries, which supported the Government of India in expanding the basket of contraceptive choices, contributing to meet India’s FP 2020 commitments and to universal access to quality contraceptive services. Quality along with respectful client-centered care was the cornerstone of the program.” 

 98 - 101

2 It is always challenging to evaluate a particular intervention and program as multiple programs and interventions with similar aims and objectives runs simultaneously. How did you ensure whether no interventions with similar objectives under NRHM, state health departments or any other organization overlap with the goals of this study?

 Thank you for your interesting comments. MCS program worked very closely with state health departments of both the states for implementing the described interventions during the program period. This study was also an integral part of the overall program and was conducted in coordination with the state health departments. To the best of our knowledge, there were no other programs or interventions with similar objectives that were implemented at the study sites during the course of this study. Also, the study sites did include a matched comparison arm to account for secular trends. We have added a few lines to the manuscript for including this information. 

Added lines: “Both the program and the study were conducted in close coordination with the state health departments, and there were no other quality improvement interventions with similar objectives implemented at the study sites, during course of the study.”

 499-502

3 Many of the existing literature and researches suggest better training to service providers and maintaining safety checklist ensures good quality service delivery in family planning services. This study showed that providing better training and maintaining safety checklist not always guarantee good service delivery, especially in the case of pre-operative assessment and surgery. In this context, authors could specifically suggest what could be done further to improve the quality of sterilization services in the country.

 Thank you for the feedback. We have now added a few suggestions in the conclusion section for improving quality of female sterilization services. These include inclusion of respectful care as an essential aspect of family planning service provision, incorporation of behavior change strategies in clinical trainings and reduced turn-over of trained staff.

Added text: “Utilization of behavior change models as part of the intervention might have helped improve practices around infection prevention and control. In addition, inclusion of respectful care as an essential aspect of family planning service provision and measures to ensure reduced turn-over of trained staff can further strengthen adherence to best-practices and ensure sustainability of quality improvement initiatives such as MCSP.” 528- 532

4 1. Quasi-experimental designs are subject to a variety of selection related threats such as:

• Selection-history threat (the intervention and comparison groups being differentially impacted by extraneous or historical events),

• Selection-regression threat (the intervention and comparison groups regressing toward the mean between baseline and end-line at different rates)

• Selection-instrumentation threat (the intervention and comparison groups responding differently to the measurement)

• Selection-testing (the intervention and comparison groups responding differently to the baseline)

• Selection-mortality (the intervention and comparison groups demonstrating differential dropout rates. In this study there was a shortfall in the number of observations completed in comparison facilities at end-line)

What measures were adopted in this study to overcome those threats mentioned above?

 Thank you for raising this very important point. Below are the point wise responses – 

1. Selection history threat – To mitigate this threat, we did an a priori matching of the facilities based on three criteria – location of the facility, delivery load as well as load of female sterilization cases in the three months prior to baseline data collection. Also, the study was conducted in close coordination with health departments of both the states, and to the best of our knowledge there were no other programs or interventions (extraneous events) with similar objectives that were implemented at the study sites during the course of this study

2. Selection regression threat – An a priori matching of the facilities and the fact that baseline scores on most practices are comparable between the two groups, would help mitigate this threat. 

3. Selection instrumentation threat – We used a standard observation checklist to complete observations across facilities of both the study arms. Data collectors underwent a standardized 3 day training by the same set of trainers before each round of data collection. We were also able to retain 70% data collectors from the baseline to endline. All these measures, we believe have helped mitigate the selection instrumentation threat. 

4. Selection testing threat – The results of this study were shared with the respective state health departments only after completion of the endline assessment. Also, the observers were trained to make their observations in an as unobtrusive manner as possible, so as to minimize the Hawthorne effect. These measures, we believe may have mitigated the said threat. 

5. Selection mortality – We replaced three facilities in the comparison arm, which were no longer performing minilap surgeries using the same criteria for matching and ensuring that there were no other quality improvement interventions with similar objectives being implemented at their site. This we believe may have mitigated the said threat. 

We have included points relevant to each of these different threats primarily in the strengths and limitations section (last paragraph of the discussion section) of our manuscript as well as other relevant sections. 494 - 517

---

## [Editor Report · Decision Letter 1]

3 Dec 2020

Effectiveness of a quality improvement intervention to increase adherence to key practices during female sterilization services in Chhattisgarh and Odisha states of India

PONE-D-20-30411R1

Dear Dr. Chhibber,

We’re pleased to inform you that your manuscript has been judged scientifically suitable for publication and will be formally accepted for publication once it meets all outstanding technical requirements.

Kind regards,

Vijayaprasad Gopichandran

Academic Editor

PLOS ONE
---

## [Editor Report · Acceptance letter]

14 Dec 2020

PONE-D-20-30411R1 

Effectiveness of a quality improvement intervention to increase adherence to key practices during female sterilization services in Chhattisgarh and Odisha states of India 

Dear Dr. Chhibber:

I'm pleased to inform you that your manuscript has been deemed suitable for publication in PLOS ONE. Congratulations! Your manuscript is now with our production department. 

Kind regards, 

on behalf of

Dr. Vijayaprasad Gopichandran 

Academic Editor

PLOS ONE